# A Novel Synthetic Dihydroindeno[1,2-b] Indole Derivative (LS-2-3j) Reverses ABCB1- and ABCG2-Mediated Multidrug Resistance in Cancer Cells

**DOI:** 10.3390/molecules23123264

**Published:** 2018-12-10

**Authors:** Chao Guo, Fangyuan Liu, Jie Qi, Jiahui Ma, Shiqi Lin, Caiyun Zhang, Qian Zhang, Hangyu Zhang, Rong Lu, Xia Li

**Affiliations:** 1School of Ocean, Shandong University, Weihai 264209, China; super70732635@163.com (C.G.); fangyuan617@outlook.com (F.L.); 18769742827@163.com (J.Q.); sdumjh@hotmail.com (J.M.); lsqsd@outlook.com (S.L.); caiyun617@outlook.com (C.Z.); zhangqianzq@sdu.edu.cn (Q.Z.); 2Department of Biomedical Engineering, Faculty of Electronic Information and Electrical Engineering, Dalian University of Technology, Dalian 116024, China; hangyuz@dlut.edu.cn; 3Research Center for the Control Engineering of Translational Precision Medicine, Dalian University of Technology, Dalian 116024, China; 4The Key Laboratory of Chemistry for Natural Product of Guizhou Province, Chinese Academy of Science, Guiyang 550002, China

**Keywords:** indole derivative, multidrug resistance, K562/A02 cell line, MCF-7/ABCG2 cell line, ABCB1, ABCG2

## Abstract

10-oxo-5-(3-(pyrrolidin-1-yl) propyl)-5,10-dihydroindeno [1,2-b] indol-9-yl propionate (LS-2-3j) is a new chemically synthesized indole compound and some related analogues are known to be inhibitors (such as alectinib and Ko143) of ATP-binding cassette (ABC) transporters, especially the ABC transporter subfamily B member 1 (ABCB1) and the ABC transporter subfamily G member 2 (ABCG2). This study aimed to evaluate the multidrug resistance (MDR) reversal effects and associated mechanisms of LS-2-3j in drug-resistant cancer cells. The inhibition of cell proliferation in tested agents was evaluated by the 3-(4,5-dimethylthiazol)-2,5-diphenyltetrazolium bromide (MTT) assay. Accumulation or efflux of chemotherapy drugs was analyzed by flow cytometry. The ATPase activity was measured using an ATPase activity assay kit. The mRNA transcripts and protein expression levels were detected by real-time PCR and Western blot, respectively. In this connection, LS-2-3j significantly enhanced the activity of chemotherapeutic drugs in MDR cells and could significantly increase the intracellular accumulation of doxorubicin (DOX) and mitoxantrone (MITX) by inhibiting the function of the efflux pumps in ABCB1- or ABCG2-overexpressing cells. Furthermore, reduced ATPase activity, mRNA transcription, and protein expression levels of ABCB1 and ABCG2 were observed in a concentration dependent manner in MDR cancer cells.

## 1. Introduction

Multidrug resistance (MDR) means that cancer cells not only develop resistance to a type of anticancer drug, but also cross-resistance to other anticancer drugs with different structures and different mechanisms of action. MDR to chemotherapeutic drugs is a serious problem that, in many cases, leads to cancer treatment failure. There are many mechanisms leading to MDR such as DNA methylation [1], efflux transportation [2], apoptosis resistance [3], topoisomerase II regulation [1], Glutathione (GSH) detoxification [4] and epithelial–mesenchymal transition [5]. Among these factors, the most prominent cause of MDR is the overexpression of the ATP-binding cassette (ABC) superfamily of transporters, which can transport intracellular anticancer drugs out of cells, leading to decreased drug accumulation in cells [6].

Among the family of transmembrane proteins, ABC transporters occupy a large part. They contain 49 transporters from seven subfamilies, ABC-A to -G, based on sequence similarities [7]. Among these transporters, the ABC transporter subfamily B member 1 (ABCB1) and the ABC transporter subfamily G member 2 (ABCG2) are known to play an important role in mediating multidrug resistance in cancer cells [8,9]. Studies have shown that ABCB1 and ABCG2 are overexpressed in many multidrug resistant cancer cells including ovarian, colon, lung, breast, and melanoma cancers [10,11,12]. The substrates of ABCB1 include doxorubicin (DOX), vincristine (VCR), and paclitaxel, while flavopiridols, mitoxantrone (MITX), nucleoside analogues, and organic anion conjugates are known to be the substrates of ABCG2 [13]. Furthermore, ABCB1 and ABCG2 can co-exist on the surface of many multidrug resistant cancer cells [14]. Much effort has been put into the clinical development of useful MDR reversal agents. Thus, it is significant in clinical studies to explore common inhibitors of ABCB1 and ABCG2. LS-2-3j is a novel indole-based small molecule synthesized by our chemistry group. Since LS-2-3j has some analogues that are known to be ABCB1 and ABCG2 inhibitors (alectinib and Ko143, etc.) [15,16,17], we investigated MDR reversal by this compound in this study. We found that LS-2-3j inhibited both ABCB1 and ABCG2 in multidrug resistant cells in vitro and the underlying mechanisms were further investigated.

## 2. Results

### 2.1. Cell Cytotoxity Assay

The 3-(4,5-dimethylthiazol)-2,5-diphenyltetrazolium bromide (MTT) colorimetric assay was performed to assess the cytotoxic effects of tested drugs in cancer cells, MDR cells, and non-cancerous cells. The IC50 (cell inhibition ratio was 50%) values of LS-2-3j are 3.188 ± 0.153, 2.434 ± 0.208, 4.019 ± 0.189, 4.580 ± 0.880, 4.814 ± 0.494, 4.053 ± 0.680, 2.758 ± 0.230, 7.355 ± 0.357, 6.800 ± 0.723, 10.835 ± 0.215, and 12.183 ± 0.2235 μmol/L for MCF-7, MCF-7/Adr, MCF-7/ABCG2, K562, K562/A02, KB, KB/VCR, 293T, 293T/ABCG2, BEAS-2B, and LO2 cells, respectively (Figure 1A–F). Compared with cancer cells, LS-2-3j is less toxic to 293T, BEAS-2B, and LO2 normal cells. Based on the results of compound cytotoxicity, 0.25, 0.5, and 1 μmol/L were chosen as the three concentrations for further investigation into the combination of LS-2-3j and chemotherapeutic agents. There is no significant cell cytotoxicity for the above three concentrations in all cell lines used in the MDR reversal study.

### 2.2. LS-2-3j Enhances the Sensitivity of Chemotherapeutic Agents

The activity of LS-2-3j to reverse the resistance of MDR cells to chemotherapeutic agents was investigated by an MTT assay and the results are shown in Table 1 and Table 2. Pretreatment with LS-2-3j exhibits a strong potency to reverse DOX or VCR in various MDR cells (Table 1). Moreover, in a concentration-dependent manner, LS-2-3j increases MITX sensitivity to the MCF-7/Adr cells and stably transfects MCF-7/ABCG2 and 293T/ABCG2 cells (Table 2). Reversal fold (RF) values of 1 μmol/L LS-2-3j are comparable to verapamil (VRP) and fumitremorgin C (FTC). In contrast, no such effects are observed in their parental MCF-7, K562, KB, and 293T cells. Moreover, it should be noted that LS-2-3j could not reverse resistance to non-ABCB1 or non-ABCG2 substrates (cisplatin) in MDR cells. These results suggest that LS-2-3j enhances the sensitivity of conventional anticancer drugs that are substrates of ABCB1 and ABCG2.

### 2.3. LS-2-3j Enhances the Accumulation of DOX and MITX

The effects of LS-2-3j on enhancing the sensitivity of ABCB1- and ABCG2-overexpressing cells to conventional anti-cancer drugs were further detected by the intracellular DOX- and MITX-associated mean fluorescence intensity (MFI) using flow cytometry (Figure 2). Compared with the parental sensitive cells, the intracellular accumulation levels of DOX and MITX are lower in MDR cells (Figure 2C,F). Pretreatment with LS-2-3j markedly increases the intracellular accumulation of DOX or MITX in a concentration-dependent manner for K562/A02 or MCF-7/ABCG2 cells; with an MFI fold change ranging from 1.830 to 4.026 in the K562/A02 cells (Figure 2B,C), and 1.307 to 2.721 in MCF-7/ABCG2 cells (Figure 2E,F). In contrast, the DOX or MITX concentration in the corresponding parental sensitive cells remains unchanged in the presence of LS-2-3j (Figure 2A,C,D,F). These data indicate that LS-2-3j elevates the sensitivity of MDR cells toward chemotherapeutic drugs by increasing drug accumulation in the cells.

### 2.4. LS-2-3j Inhibits the Efflux of DOX and MITX

Next, we further examined the role of LS-2-3j for the outward transport function of ABCB1 and ABCG2 by measuring the time course of DOX and MITX intracellular retention. Compared with the parental K562 and MCF-7 cells, a notable decrease of DOX and MITX accumulation was monitored after 2 h in the corresponding K562/A02 and MCF7/ABCG2 cells (Figure 3). In the presence of 1 μmol/L LS-2-3j, DOX efflux is markedly suppressed in K562/A02 cells (Figure 3A,C). Similarly, intracellular MITX accumulation in ABCG2-overexpressing MCF-7/ABCG2 cells with LS-2-3j pretreatment is greater than in the untreated MCF-7/ABCG2 cells (Figure 3B,D). These results suggest that LS-2-3j can inhibit the efflux of anti-cancer drugs in MDR cells overexpressing ABCB1 and ABCG2.

### 2.5. LS-2-3j Inhibits the ATPase Activity of ABCB1 and ABCG2

Energy released by ATP hydrolysis is required for ABC transporters to pump their substrate drugs outside cells against a concentration gradient. To investigate the inhibitory function of the compound LS-2-3j on ABCB1 and ABCG2, ATPase hydrolysis ability was measured with the presence of LS-2-3j at various concentrations, from 0 to 20 μmol/L. The results show that LS-2-3j significantly inhibits the ATPase activities of ABCB1 and ABCG2 in a concentration-dependent manner (Figure 4), which suggests that LS-2-3j can inhibit the ATPase activity of ABCB1 and ABCG2 directly.

### 2.6. LS-2-3j Decreases the mRNA Expression Levels of ABCB1 and ABCG2

Since the proteins ABCB1 and ABCG2 are encoded by the genes ABCB1 and ABCG2, real-time PCR analysis was carried out to assess whether LS-2-3j modulated ABCB1 and ABCG2 mRNA levels. The results show that the mRNA levels of ABCB1 and ABCG2 significantly decrease in K562/A02 and MCF-7/Adr cells when treated with LS-2-3j for 48 h (Figure 5A,B). Furthermore, with 24 h LS-2-3j treatment, mRNA levels of ABCB1 and ABCG2 show a decreasing tendency in a dose depended manner, but no significant difference when compared with the control (Figure 5C,D). The above results suggest that LS-2-3j decreases the mRNA expression levels of ABCB1 and ABCG2 in ABCB1- or ABCG2-overexpressing MDR cells.

### 2.7. LS-2-3j Decreases the Protein Expression of ABCB1 and ABCG2

Protein expression levels of ABCB1 and ABCG2 were demonstrated with Western blot analysis. Compared with the parental K562 and MCF-7 cell lines, the protein expression levels of ABCB1 and ABCG2 are significantly higher in the K562/A02 and MCF-7/Adr cell lines. The various LS-2-3j exposures for 48 h led to a dramatic decrease in ABCB1 and ABCG2 protein expression in MDR cells (Figure 6A,B). However, with 24 h LS-2-3j treatment, protein levels of ABCB1 and ABCG2 show a decreasing tendency in a dose-dependent manner, but no significant difference when compared with the control (Figure 6C,D), which suggests that LS-2-3j has inhibitory effects on the protein levels of both ABCB1 and ABCG2.

## 3. Discussion

Multidrug resistance is one of the main causes of chemotherapy failure [18]. Previous studies have shown that there are many strategies to overcome MDR [19]. Chemical sensitizers for drug-resistant cancer cells have been developed such as calcium channel blockers (CCB) [20], cyclosporine A and its derivatives [21], anti-estrogen [22], enzyme inhibiters [23], and so on. Additionally, genetic engineering has been applied to reverse multidrug resistance using antisense oligodeoxyribonudeotides (AOD) [24], RNA interference (RNAi) [25] and other techniques. Among these, the development of reversal agents for ABC transporters remains an important research direction [26]. Repositioning drugs as ABC transporter inhibitors is of great important to overcome multidrug resistance in clinical therapies. Thus, inhibitors targeting both ABCB1 and ABCG2 transporters have broad prospects for clinical chemotherapy.

LS-2-3j is a novel chemically synthesized indole compound with a structure similar to known ATP-binding cassette (ABC) transporter inhibitors, e.g., alectinib and Ko143. Alectinib is a dual ABCB1 and ABCG2 inhibitor, while Ko143 is a selective ABCG2 inhibitor. These three compounds all have four linked rings that hold a nitrogen and oxygen. In this case, we performed molecular experiments to explore the reversal of MDR by LS-2-3j. Here, MTT assays show that LS-2-3j, at its non-toxic concentration, markedly potentiated the efficacy of substrate chemotherapeutic agents such as DOX, VCR, and MITX in ABCB1- or ABCG2-overexpressing cells. We observe more than 80% cell survival at this concentration of LS-2-3j, with no effect on the chemotherapeutic sensitive cells. Since the chemotherapeutic drug cisplatin is neither a substrate for ABCB1 nor for ABCG2, LS-2-3j does not increase the sensitivity of cisplatin to multidrug resistant cells. These findings suggest that ABCB1- and ABCG2-mediated MDR are involved in the ability of LS-2-3j to increase sensitivity to chemotherapeutic drugs. Three non-cancer cell lines (293T, BEAS-2B, and LO2) were tested to confirm the toxicity of LS-2-3j. LS-2-3j exhibits growth inhibitory effects in all tested cell lines. Although LS-2-3j has a potential therapeutic window with IC50 values of 3.19–4.58 μmol/L for cancer cells or MDR cells and 7.36–12.18 μmol/L for the three non-cancerous cells, developing it into a cancer therapeutic, via its MDR reversal, must be cautiously performed. As a leading compound, further structure optimization studies of LS-2-3j to decrease toxicity are warranted.

The chemotherapeutic drug accumulation results further confirm the above hypothesis. DOX is known to be a substrate of ABCB1 and ABCG2, while MITX is one of the main substrates of ABCG2 [13,27,28]. The low levels of DOX and MITX accumulation in MDR cells may account for the high IC50 values. LS-2-3j exhibits potent effects by increasing intracellular DOX and MITX uptake in ABCB1- and ABCG2-overexpressing cells, which provides clues for the mechanisms of its reversal effects. Furthermore, drug efflux was detected in ABCB1- and ABCG2-overexpressing cells. After parental sensitive cells and MDR cells were incubated with the corresponding drugs, the intracellular accumulation of DOX and MITX in MDR cells was lower than that in sensitive parental cells, and was notably increased by adding LS-2-3j to the MDR cells; LS-2-3j has no effect in sensitive parental cells. In contrast, the efflux of DOX and MITX in LS-2-3j-treated MDR cells is lower than that of untreated MDR cells. These results suggest that LS-2-3j reverses the ABCB1- and ABCG2- mediated MDR by inhibiting the transporter mediated drug efflux, thereby increasing the intracellular drug accumulation. The ABC transporter releases energy via hydrolysis of ATP to transport intracellular substances out of the cell. The results show that LS-2-3j can directly inhibit the ATPase activity of ABCB1 and ABCG2. The reversal effects of LS-2-3j on ABCB1 and ABCG2 are therefore related to its inhibition of these ABC transporters.

In the process of selecting drug-resistant cells in vitro, the high expressions of ABCB1 and ABCG2 have become important indicators of drug-resistant cells. K562/A02 cells have been shown to express ABCB1 at a high level on the plasma membrane and ABCG2 can be highly expressed in MCF-7/Adr cells. Therefore, real-time PCR and Western blot were used to detect the effect of LS-2-3j on the protein and gene expressions of ABCB1 and ABCG2. Concordantly, the results showed that there is a pronounced decrease in the mRNA level and related protein expression of ABCB1 and ABCG2 when treated with LS-2-3j after 48 h, which provides more strong evidence to explain the effects of LS-2-3j on the reversal of anticancer drug efflux in ABCB1- and ABCG2-expressing cells. However, there are still several likely complex molecular pathways associated with MDR mediated by ABC transporters, and the mechanisms that induce over-expression of ABC transporters are still unclear at present. Various intracellular signaling systems have been reported to regulate the expressions of ABC transporters. The mitogen-activated protein kinase (MAPK)/extracellular signal-regulated kinase (ERK) pathway [29] and the c-Jun N-terminal kinases (JNK) pathway [30] are reported to be involved in the regulation of ABCB1 gene expression. These two MAPK pathways have also been reported to regulate the expression of the ABCG2 gene [31,32,33]. Although the mechanism of down-regulation for LS-2-3j was not observed in this manuscript, it will be of great relevance for us to further understand such mechanisms in order to down-regulate the expression of ABCB1 and ABCG2.

Over the past three decades, researchers have put a lot of effort into exploring inhibitors of transmembrane transporters, and many inhibitors have also achieved good results on in vitro experiments. However, in clinical trials, unsatisfactory situations often occur due to unforeseen side effects or a lack of good clinical benefit. There are many reasons why transmembrane transport protein regulators fail in clinical trials. First, in addition to the transmembrane transporter family, there are many other factors that mediate multidrug resistance in tumor cells, so inhibition of transmembrane transporters alone may not achieve the desired results. Second, many tumors are capable of simultaneously expressing multiple transmembrane transporters, and many chemotherapeutic drugs may be substrates for multiple transmembrane transporters. Therefore, reversal agents against a single transmembrane protein cannot completely reverse the multidrug resistance of tumor cells. In addition, the high expression of transporters is negatively correlated with the prognosis of tumor cells. Patients with high expression of these transporters usually have a late tumor stage, are older age, and have weakened kidney function, resulting in poor prognosis and toxic side effects. For the above reasons, it is difficult to find a feasible transmembrane transport protein reversal agent in clinical practice. At present, many researchers have focused on new chemical sensitizers or new methods to specifically down-regulate the expression of multidrug resistance genes, such as small molecule inhibitors, natural drugs, RNA interference, epigenetic regulation, or signal transduction pathways [27,33,34,35,36,37].

In conclusion, LS-2-3j, a new chemically synthesized indole compound reported for the first time, has multidrug resistant reversal effects. LS-2-3j reversed MDR via inhibiting the function and down-regulating the mRNA and protein expression levels of both ABCB1 and ABCG2. LS-2-3j might be useful as a lead compound in drug discovery and development for ABCB1- or ABCG2-mediated MDR cancer treatment.

## 4. Materials and Methods

### 4.1. Synthesis and Analysis of LS-3-2j

As shown in Figure 7, 10-oxo-5-(3-(pyrrolidin-1-yl) propyl)-5,10-dihydroindeno-[1,2-b] indol-9-yl acetate (LS-2-1a) was prepared. Enamine 1 was prepared from cyclohexan-1,3-dione and amine in refluxing toluene for 3 h and used directly in the next step after evaporation. A solution of enamine derivatives 1 (2.22 g, 10.0 mmol) and ninhydrin (1.96 g, 11.0 mmol) in Ac_2_O (20 mL) was stirred at 120 °C overnight. After cooled to room temperature, the reaction mixture was diluted with ice-cold water and then extracted with EtOAc. The organic layer was then washed with brine and dried over MgSO_4_. Next, the solvent was evaporated, and the residue was purified by silica gel chromatography with Dichloromethane (DCM)/MeOH (40:1) to obtain the product LS-2-1a (2.40 g, 62.0%). If necessary, the crude product could be recrystallized in EtOH to afford a pure sample. Melting point (m.p.) 125–126 °C. ^1^H-NMR (CDCl_3_) δ: 7.37 (1H, d, *J* = 6.9 Hz), 7.28–7.11 (5H, m), 6.89 (1H, dd, *J* = 6.7, 1.6 Hz), 4.32 (2H, t, *J* = 6.8 Hz,), 2.64 (6H, d, *J* = 6.2 Hz), 2.55 (3H, s), 2.19–2.09 (2H, m), 1.84 (4H, s). ^13^C-NMR (CDCl_3_) δ: 184.06, 170.10, 158.42, 144.00, 143.84, 140.73, 134.34, 132.09, 129.85, 123.66, 123.13, 118.99, 117.63, 115.93, 113.50, 108.89, 53.62, 52.25, 43.32, 28.37, 23.42, 21.41. High-resolution-electrospray ionization-mass spectrometry (HR-ESI-MS) *m*/*z*: Calculated for C_24_H_24_N_2_O_3_ [M + H]^+^ 389.18597; found: 389.18612.

Next, 0.2 mol/L NaOMe/MeOH solution (5 mL) was added to the compound LS-2-1a (155 mg, 0.4 mmol) at room temperature for 4 h. Then the solution was evaporated, and the residue was purified by silica gel chromatography using DCM/MeOH (25:1) to afford the phenol LS-2-2a (133 mg, 95.5%). m.p. 133–135 °C. ^1^H NMR (400 MHz, CDCl_3_) δ 7.30 (dd, *J* = 9.2, 4.3 Hz, 1H), 7.18 (dd, *J* = 9.1, 7.3 Hz, 2H), 7.13–7.06 (m, 1H), 7.05–6.98 (m, 1H), 6.81–6.75 (m, 1H), 6.65–6.59 (m, 1H), 4.31–4.16 (m, 2H), 2.52–2.36 (m, 6H), 2.12–1.95 (m, 2H), 1.76 (s, 4H). ^13^C NMR (101 MHz, CDCl_3_) δ 185.79, 156.92, 149.91, 143.76, 140.55, 135.79, 132.15, 129.37, 125.22, 123.18, 119.22, 114.74, 113.15, 107.84, 103.42, 53.92, 52.27, 43.58, 28.91, 23.53. HR-ESI-MS: Calculated for C_22_H_22_N_2_O_2_ [M + H]^+^ 347.17540; found: 347.17447.

Finally, a solution of LS-2-2a (105 mg, 0.3 mmol) and propanoyl chloride (30 μL, 0.33 mmol), in pyridine (1 mL) was stirred at room temperature overnight. Then the solvent was quenched with MeOH. The residue was purified by silica gel chromatography using DCM/MeOH (40:1) to afford 116 mg of LS-2-3j (10-oxo-5-(3-(pyrrolidin-1-yl)propyl)-5,10-dihydroindeno [1,2-b] indol-9-yl propionate) in 96.2% yield. m.p. 122–123 °C. ^1^H-NMR (CDCl_3_) δ: 7.34–7.27 (1H, m), 7.24–7.08 (5H, m), 6.89–6.84 (1H, m), 4.23 (2H, dd, *J* = 12.6, 5.8 Hz,), 2.99–2.79 (8H, m), 2.30–2.17 (2H, m), 1.95 (4H, s), 1.36 (3H, t, *J* = 7.5 Hz). ^13^C-NMR (CDCl_3_) δ: 183.81, 173.60, 157.97, 144.12, 143.59, 140.28, 133.86, 132.33, 129.89, 123.92, 123.12, 118.96, 117.93, 116.10, 113.67, 108.73, 53.78, 52.35, 42.80, 27.83, 27.23, 23.38, 8.96. HR-ESI-MS *m*/*z*: Calculated for C_25_H_26_N_2_O_3_ [M + H]^+^ 403.20162; found 403.20056.

### 4.2. Chemicals and Regents

Roswell Park Memorial Institute (RPMI)-1640 medium, Dulbecco’s Modified Eagle Medium (DMEM) and fetal bovine serum (FBS) were obtained from Gibco (Gaithersburg, MD, USA). MTT, DOX, MITX, VCR, FTC, VRP, cisplatin, and the PREDEASY™ ATPase Kit were purchased from Sigma Corp. (St. Louis, MO, USA). Geneticin (G418) was purchased from Solarbio Science and Technology Co., Beijing, China. Monoclonal antibodies against glyceraldehyde-3-phosphate dehydrogenase (GAPDH), ABCB1, and ABCG2 were obtained from Santa Crus Biotechnology (Dallas, TX, USA). RNeasy mini kit and Rever Tra Ace qPCR RT kit were purchased from QIAGEN (Hilden, Germany) and Toyobo (Osaka Prefecture, Japan), respectively. LS-2-3j, MTT, DOX, MITX, VCR, FTC, VRP, and cisplatin were dissolved in dimethylsulfoxide (DMSO) as a 10 mmol/L solution, stored at −20 °C.

### 4.3. Cell Lines and Cell Culture

Human breast adenocarcinoma cell line MCF-7 and its DOX selected ABCB1- and ABCG2-overexpressing derivative MCF-7/Adr cell line, the human myelogenous leukemia cell line K562 and its DOX selected ABCB1-overexpressing derivative K562/A02 cell line, the human oral epidermoid carcinoma cell line KB and its VCR selected ABCB1-overexpressing derivative KB/VCR cell line, the human kidney cell line 293T, the human bronchial epithelial cell line BEAS-2B, and the normal human hepatic cell line LO2 were all used in this study. All cell lines were purchased from the Shanghai Institute for Biological Sciences (SIBS, Shanghai, China). MCF-7/ABCG2 and 293T/ABCG2 cell lines were established by selection with G418 after transfecting MCF-7 and 293T with the pcDNA3.1 vector containing the full length ABCG2 gene sequence, and they were all cultured in medium with 800 μg/mL of G418. All cell lines were cultured in RPMI-1640 or DMEM with 10% (*v*/*v*) FBS at 37 °C in a humidified atmosphere of 5% CO_2_. The cells were incubated for two weeks under the above drug-free conditions for the following experiments.

### 4.4. MTT Assay

The MTT assay was used to assess the cytotoxicity of the compound LS-2-3j and its ability to potentiate the cytotoxicity of DOX, VCR, MITX, and cisplatin. Briefly, cells growing in the logarithmic phase were seeded into 96-well plates at concentrations of 2 × 10^4^ per well. Then, cells were incubated for 24 h until they were completely adherent. Next, a range of different concentrations of conventional anticancer drugs (DOX, MITX, VCR, and cisplatin) with or without a reversal compound were added and incubated for 48 h. The IC50 and RF values were calculated from the results.

### 4.5. Drug Intracellular Accumulation Assay

The effect of LS-2-3j on the accumulation of DOX and MITX was measured by flow cytometry (Becton Dickinson FACScan, San Jose, CA, USA). Briefly, the cells were incubated in 6-well plates and allowed to be adherent overnight. Then, the cells were exposed to a range of different concentrations of the test compound LS-2-3j (0, 0.25, 0.5 and 1 μmol/L). After 1 h, DOX (5 μmol/L) or MITX (5 μmol/L) were added into the medium and incubated for another 1 h. Cells were then digested, collected, and washed three times with ice-cold phosphate-buffered saline (PBS) buffer. Finally, cells were re-suspended in 400 μL of ice-cold PBS buffer and detected by flow cytometric analysis. The degree of intracellular drug accumulation was finally presented by the MFI. VRP and FTC were used as the positive control in the above experiments.

### 4.6. Drug Efflux Assay

In the efflux study, the cells were incubated in 6-well plates for 24 h to be adherent. Later, cells were incubated with medium containing DOX (5 μmol/L) or MITX (5 μmol/L) for 1 h at 37 °C, then the cells were washed three times with ice-cold PBS buffer and subsequently maintained at 37 °C with culture media without MITX or DOX in the presence or absence of 1 μmol/L LS-2-3j for 0, 30, 60, 90 and 120 min. After that, cells were collected and washed three times with ice-cold PBS buffer and re-suspended with 400 μL PBS buffer for immediate flow cytometric analysis. VRP and FTC were used as the positive controls.

### 4.7. ATPase Activity Assay

The ATPase activities influenced by LS-2-3j were measured by the MDR1/P-gp (multidrug resistance protein 1/P-glycoprotein) PREDEASY ATPase Kit (Sigma Corp.) and the BCRP (breast cancer resistance protein) M PREDEASY ATPase Kit (Sigma Corp.) with modified protocols. Briefly, the ABCB1- and ABCG2- membrane suspension was thawed and diluted. Some of the ATPase activities in the membranes are Na_3_VO_4_ insensitive. As the ABCB1- and ABCG2-specific activity is Na_3_VO_4_ sensitive, Na_3_VO_4_ was used as an ATPase inhibitor. The membranes were exposed to the test compound LS-2-3j (0–20 μmol/L) for 5 min. Then the ATPase reactions were initiated by adding 5mM Mg^2+^ ATP. After being incubated at 37 °C for 40 min, luminescence signals of Pi were initiated and measured according to the instructions of the PREDEASY ATPase Kit.

### 4.8. RNA Extraction and Real-Time PCR

Real-time PCR was carried out to analyze the mRNA levels of ABCB1 and ABCG2. Cells were incubated in 6-well plates and allowed to become adherent overnight. Then, the cells were exposed to the test compound LS-2-3j (0, 0.25, 0.5 and 1 μmol/L) for 24 or 48 h. After that, RNA was extracted by the RNeasy Mini Kit (QIAGEN) according to the manufacturer’s instructions. Then, cDNA was synthesized through reverse transcription using the Rever Tra Ace qPCR RT kit (QIAGEN) and according to the related instructions. PCR amplification was performed in an 8-tube strip format (Axygen, Union City, CA, USA) in triplicate. For each reaction, 1 μL SYBR Green PCR Master Mix (1×), 1 μL forward and reverse primer (10 μmol/L), and 1 μL template cDNA were added to a final volume of 20 μL with diethyl pyrocarbonate (DEPC) water. Primers were based on the ABCB1 gene (forward: 5′-CCATGGAGAAGGCTGGGG-3′, reverse: 5′-CCAAGTTGTCATGGATGACC-3′), the ABCG2 gene (forward: 5′-GCCATAGCAGCAGGTCAG-3′, reverse: 5′-AGCCGTAAATCCATATCGTG-3′), and the GAPDH gene (forward: 5′-GACCTGACCTGCCGTCTA-3′, reverse: 5′-AGGAGTGGGTGTCGCTGT-3′). Reactions were performed for 45 cycles of sequential denaturation at (95 °C, 2 min), annealing (58 °C, 15 s) and extension (72 °C, 20 s). All primers in the above experiments were synthesized by Rui Bochenko Biological Technology Co., Ltd. (Qingdao, China).

### 4.9. Western Blot Analysis

Western blot analysis was used to analyze the protein expression levels of ABCB1 and ABCG2. Cells were incubated with LS-2-3j as described in Section 4.8 and collected and washed three times with ice-cold PBS buffer, centrifuged at 1000× *g* for 5 min. After extracted and separated by 12% sodium dodecyl sulfate (SDS)-polyacrylamide gel electrophoresis, proteins were transferred to nitrocellulose filter membranes and blocked for 1 h in tris-buffered saline and Tween 20 buffer (TBS-T buffer: 10 mmol/L Tris-HCl, 150 mmol/L NaCl, and 0.1% (*v*/*v*) Tween-20; pH 7.8) with 5% non-fat milk. The membranes were then incubated with primary antibodies against ABCB1, ABCG2, and GAPDH at 4 °C overnight. After washing four times with TBS-T buffer for 5 min, the membranes were incubated with horseradish peroxidase (HRP)-conjugated goat anti-rabbit IgG secondary antibody (Jackson, West Grove, PA, USA) for 1 h at room temperature. Finally, the proteins on the membranes were visualized using developer, fixer, and X-ray film. Protein expression levels were quantified by densitometry using ImageJ (National Institutes of Health, Bethesda, MD, USA) and GraphPad Prism 5.01 software (GraphPad, San Diego, CA, USA).

### 4.10. Statistical Analysis

All experiments were performed at least three times. All data were represented as the mean ± SD and analyzed using ANOVA, followed by Tukey’s multiple comparison test (conducted using GraphPad Prism 5.01). Significant difference was considered to be at *p* < 0.05.

## Figures and Tables

**Figure 1 molecules-23-03264-f001:**
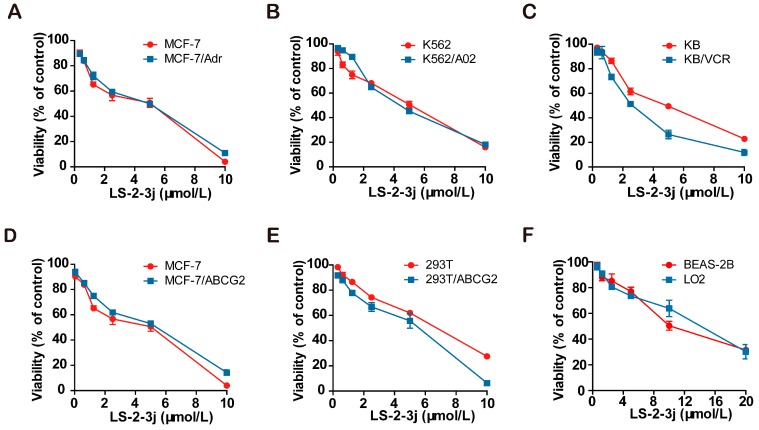
Cytotoxicity of LS-2-3j alone in drug-sensitive and drug-resistant cell lines. An The 3-(4,5-dimethylthiazol)-2,5-diphenyltetrazolium bromide (MTT) cytotoxicity assay was assessed in ABCB1- or ABCG2-overexpressing cells, sensitive parental cancer cells, and non-cancerous cells: (**A**) MCF-7 and ABCB1- and ABCG2-overexpressing MCF-7/Adr cells; (**B**) K562 and ABCB1-overexpressing K562/A02 cells; (**C**) KB and ABCB1-overexpressing KB/VCR cells; (**D**) MCF-7 and ABCG2-overexpressing MCF-7/ABCG2 cells; (**E**) 293T and ABCG2-overexpressing 293T/ABCG2 cells; (**F**) BEAS-2B and LO2 non-cancer cells. All cells were treated with a range of different concentrations of LS-2-3j for 48 h. Data are expressed as mean ± SD for three independent experiments.

**Figure 2 molecules-23-03264-f002:**
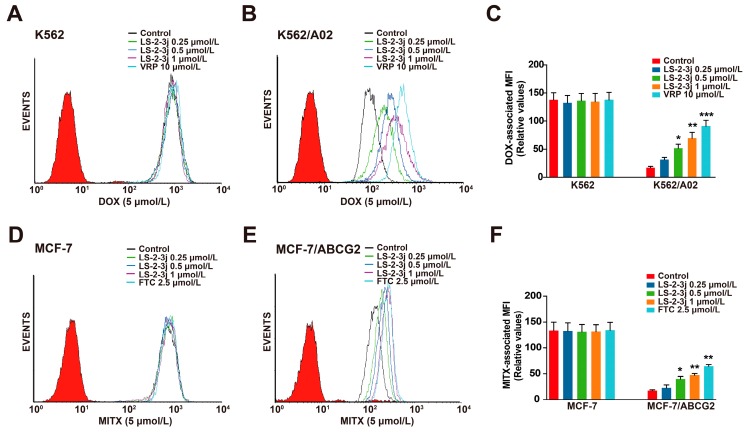
Effect of LS-2-3j on the intracellular accumulation of DOX and MITX in K562 (**A**), K562/A02 (**B**), MCF-7 (**D**), and MCF-7/ABCG2 cells (**E**). The cells were exposed to DOX (5 μmol/L) and MITX (5 μmol/L) in the absence or presence of different concentrations of LS-2-3j for 1 h. (**C**,**F**) The DOX- and MITX-associated mean fluorescence intensity (MFI) in K562/A02, MCF-7/ABCG2 cells, and their parental cells was measured by flow cytometric analysis. The results are presented as fold change to the control group. Each bar represents the mean ± SD of three independent experiments. * *p* < 0.05, ** *p* < 0.01, *** *p* < 0.001 vs. the control group.

**Figure 3 molecules-23-03264-f003:**
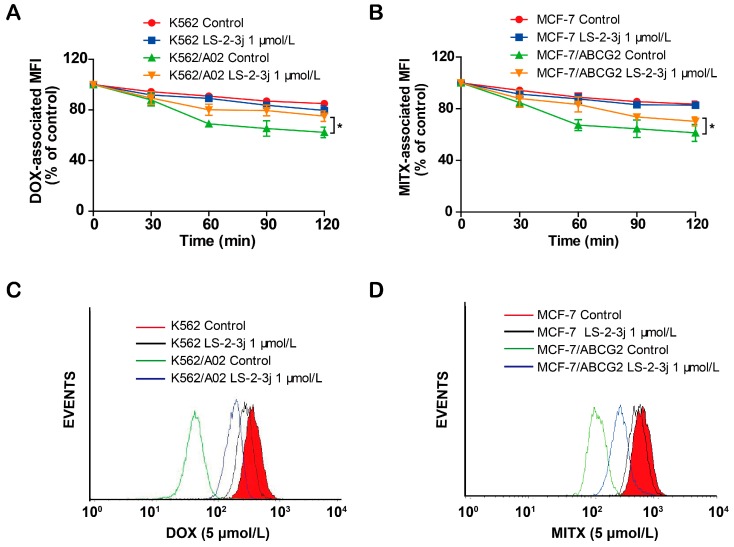
LS-2-3j inhibited the efflux of DOX and MITX. (**A**,**B**) The effect of LS-2-3j on the efflux of DOX and MITX in K562, K562/A02, MCF-7, and MCF-7/ABCG2 cells. (**C**,**D**) The corresponding flow cytometric analysis peak at the 120 min time point for various test compounds. Cells were exposed to DOX (5 μmol/L) or MITX (5 μmol/L) for 60 min and then incubated with LS-2-3j (1 μmol/L) for 0, 30, 60, 90, or 120 min. The DOX- and MITX-associated MFI was examined by flow cytometry. Data are expressed as mean ± SD of three independent experiments.

**Figure 4 molecules-23-03264-f004:**
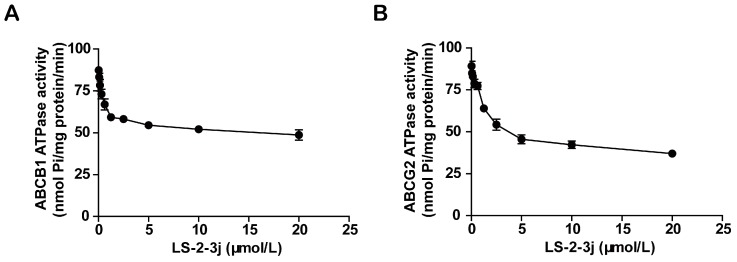
Effect of LS-2-3j on orthovanadate (Vi)-sensitive ABCB1 (**A**) and ABCG2 (**B**) ATPase activity with increased LS-2-3j concentration (0–20 μmol/L). Each bar represents the mean ± SD of three independent experiments.

**Figure 5 molecules-23-03264-f005:**
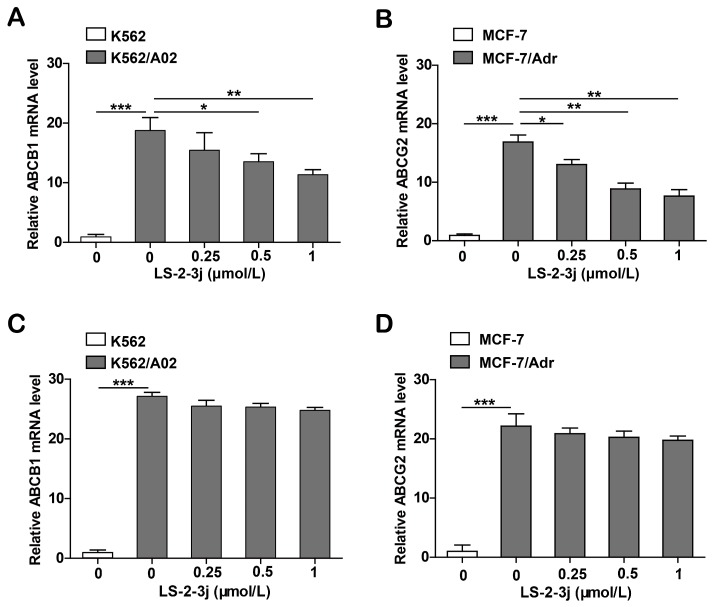
Analysis of LS-2-3j treatment on ABCB1 and ABCG2 mRNA levels in K562, K562/A02, MCF-7, and MCF-7/Adr cells. (**A**,**B**) The ABCB1 and ABCG2 mRNA levels following treatment with LS-2-3j after 48 h. (**C**,**D**) The ABCB1 and ABCG2 mRNA levels following treatment with LS-2-3j after 24 h. ABCB1 and ABCG2 mRNA levels are induced with a notable decrease in a dose-dependent manner at 48 h. mRNA levels of ABCB1 and ABCG2 have a decreasing tendency in a dose-dependent manner at 24 h. The results are presented as fold change compared to the control group of K562/A02 or MCF-7/Adr. Each bar represents the mean ± SD of three independent experiments. * *p* < 0.05, ** *p* < 0.01, *** *p* < 0.001 vs. the K562/A02 or MCF-7/ABCG2 control group.

**Figure 6 molecules-23-03264-f006:**
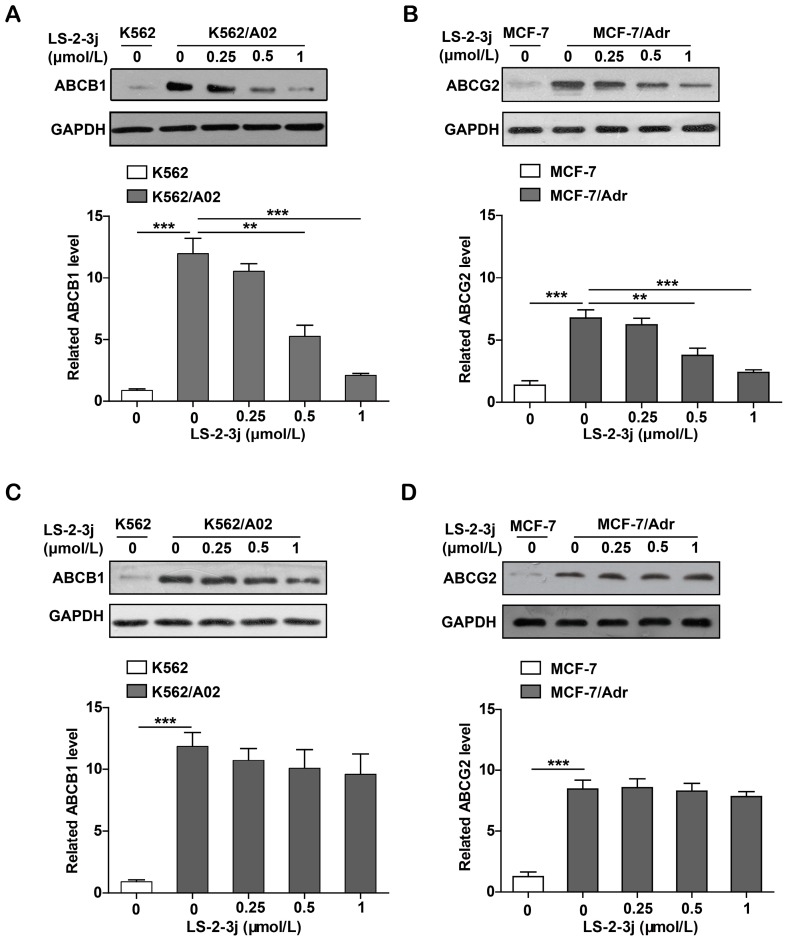
Effect of LS-2-3j on the expression levels of ABCB1 or ABCG2 proteins in K562, K562/A02, MCF-7, and MCF-7/Adr cells. (**A**,**B**) Effect of LS-2-3j on the expression levels of ABCB1 or ABCG2 proteins at 48 h. (**C**,**D**) Effect of LS-2-3j on the expression levels of ABCB1 or ABCG2 proteins at 24 h. The protein levels of ABCB1 and ABCG2 in cells after 0, 0.25, 0.5, and 1 μmol/L LS-2-3j stimulation for 24 or 48 h were measured by Western blot analysis. Protein levels of ABCB1 and ABCG2 are induced with a significant decrease at 48 h. However, there is no significant differences at 24 h. The corresponding histograms showing the quantification of densitometry are presented. The results are presented as fold change to the control group of K562/A02 or MCF-7/Adr. Glyceraldehyde-3-phosphate dehydrogenase (GAPDH) is the internal control protein in the figure. Each bar represents the mean ± SD of three independent experiments. * *p* < 0.05, ** *p* < 0.01, *** *p* < 0.001 vs. the K562/A02 or MCF-7/Adr control group.

**Figure 7 molecules-23-03264-f007:**
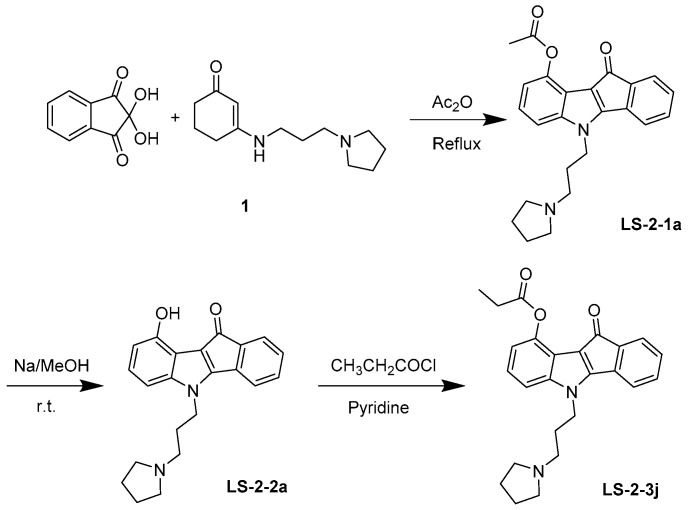
Synthesis and chemical structure of LS-2-3j.

**Table 1 molecules-23-03264-t001:** LS-2-3j reverses ABCB1-mediated drug resistance in ABCB1-overexpressing cell lines.

**Treatment**	**IC50 ± SD (μmol/L) (Reversal Fold, RF)**			
	**MCF-7**		**MCF-7/Adr**	
Doxorubicin (DOX)	0.298 ± 0.008	(1.000)	3.506 ± 0.637	(1.000)
+ LS-2-3j 0.25 μmol/L	0.313 ± 0.019	(0.953)	0.827 ± 0.128 *	(4.241)
+ LS-2-3j 0.5 μmol/L	0.303 ± 0.012	(0.986)	0.471 ± 0.072 **	(7.438)
+ LS-2-3j 1 μmol/L	0.300 ± 0.005	(0.995)	0.353 ± 0.031 **	(9.937)
+ VRP 10 μmol/L	0.317 ± 0.017	(0.941)	0.274 ± 0.057 **	(12.812)
Cisplatin	7.004 ± 0.662	(1.000)	4.363 ± 0.189	(1.000)
+ LS-2-3j 1 μmol/L	7.316 ± 0.913	(0.957)	4.581 ± 0.234	(0.952)
**Treatment**	**IC50 ± SD (μmol/L) (Reversal Fold, RF)**			
	**K562**		**K562/A02**	
DOX	0.118 ± 0.001	(1.000)	13.317 ± 0.840	(1.000)
+ LS-2-3j 0.25 μmol/L	0.118 ± 0.010	(0.997)	4.947 ± 0.580 *	(2.692)
+ LS-2-3j 0.5 μmol/L	0.127 ± 0.009	(0.925)	3.182 ± 0.162 *	(4.185)
+ LS-2-3j 1 μmol/L	0.126 ± 0.010	(0.934)	1.676 ± 0.269 **	(7.944)
+ VRP 10 μmol/L	0.122 ± 0.007	(0.962)	1.180 ± 0.160 **	(11.282)
Cisplatin	2.747 ± 0.227	(1.000)	3.768 ± 0.124	(1.000)
+ LS-2-3j 1 μmol/L	2.223 ± 0.124	(1.236)	3.577 ± 0.314	(1.053)
	**KB**		**KB/VCR**	
Vincristine (VCR)	0.010 ± 0.001	(1.000)	1.126 ± 0.134	(1.000)
+ LS-2-3j 0.25 μmol/L	0.010 ± 0.010	(1.030)	0.589 ± 0.114 *	(1.913)
+ LS-2-3j 0.5 μmol/L	0.011 ± 0.012	(0.955)	0.291 ± 0.045 *	(3.874)
+ LS-2-3j 1 μmol/L	0.011 ± 0.030	(0.939)	0.150 ± 0.022 **	(7.531)
+ VRP 10 μmol/L	0.011 ± 0.001	(0.935)	0.115 ± 0.018 **	(9.808)
Cisplatin	0.705 ± 0.263	(1.000)	1.183 ± 0.140	(1.000)
+ LS-2-3j 1 μmol/L	0.703 ± 0.147	(1.002)	1.213 ± 0.118	(0.975)

Data are expressed as mean ± SD of three independent experiments. * *p* < 0.05, ** *p* < 0.01 vs. the 0 μmol/L LS-2-3j group.

**Table 2 molecules-23-03264-t002:** LS-2-3j reverses ABCG2-mediated drug resistance in ABCG2-overexpressing cell lines.

Treatment	IC50 ± SD (μmol/L) (RF)			
	MCF-7		MCF-7/ABCG2	
Mitoxantrone (MITX)	0.425 ± 0.014	(1.000)	3.550 ± 0.054	(1.000)
+ LS-2-3j 0.25 μmol/L	0.434 ± 0.035	(0.980)	1.639 ± 0.177 *	(2.236)
+ LS-2-3j 0.5 μmol/L	0.462 ± 0.033	(0.921)	0.755 ± 0.021 *	(4.770)
+ LS-2-3j 1 μmol/L	0.436 ± 0.012	(0.975)	0.487 ± 0.050 **	(7.479)
+ Fumitremorgin C (FTC) 2.5 μmol/L	0.444 ± 0.017	(0.958)	0.372 ± 0.023 **	(8.793)
Cisplatin	7.004 ± 0.662	(1.000)	5.478 ± 0.882	(1.000)
+ LS-2-3j 1 μmol/L	7.316 ± 0.913	(0.957)	5.248 ± 0.142	(1.044)
	**MCF-7**		**MCF-7/Adr**	
MITX	0.425 ± 0.014	(1.000)	39.242 ± 0.035	(1.000)
+ LS-2-3j 0.25 μmol/L	0.434 ± 0.035	(0.980)	30.141 ± 0.264	(1.302)
+ LS-2-3j 0.5 μmol/L	0.462 ± 0.033	(0.921)	13.772 ± 0.210 *	(2.849)
+ LS-2-3j 1 μmol/L	0.436 ± 0.012	(0.975)	8.253 ± 0.805 **	(4.755)
+ FTC 2.5 μmol/L	0.444 ± 0.017	(0.958)	4.891 ± 0.337 **	(8.023)
Cisplatin	7.004 ± 0.662	(1.000)	8.995 ± 0.364	(1.000)
+ LS-2-3j 1 μmol/L	7.316 ± 0.913	(0.957)	8.784 ± 0.963	(1.024)
	**293T**		**293T/ABCG2**	
MITX	0.187 ± 0.013	(1.000)	4.253 ± 0.200	(1.000)
+ LS-2-3j 0.25 μmol/L	0.200 ± 0.005	(0.934)	1.739 ± 0.056 *	(2.446)
+ LS-2-3j 0.5 μmol/L	0.194 ± 0.004	(0.963)	1.205 ± 0.065 *	(3.530)
+ LS-2-3j 1 μmol/L	0.183 ± 0.008	(1.020)	0.707 ± 0.051 **	(6.015)
+ FTC 2.5 μmol/L	0.189 ± 0.019	(0.985)	0.466 ± 0.016 **	(9.120)
Cisplatin	1.983 ± 0.132	(1.000)	1.411 ± 0.197	(1.000)
+ LS-2-3j 1 μmol/L	1.970 ± 0.184	(1.006)	1.551 ± 0.189	(0.910)

Data are expressed as mean ± SD of three independent experiments. * *p* < 0.05, ** *p* < 0.01 vs. the 0 μmol/L LS-2-3j group.

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
