# Peer review of "A Novel Synthetic Dihydroindeno[1,2-b] Indole Derivative (LS-2-3j) Reverses ABCB1- and ABCG2-Mediated Multidrug Resistance in Cancer Cells"

_molecules, 2018, doi:10.3390/molecules23123264_

Round 1
Reviewer 1 Report
The manuscript by Chao Guo ad coll. evaluated the efficacy of the new indole-derivative LS-2-3j in reversing multidrug resistance mediated by ABCB1 and ABCG2 in three highly resistant cell lines, in their sensitive parental counterpart and in three “non-cancer” cell lines. They observed high activity of the compound in increasing the cell toxicity as well as cell uptake and efflux inhibition of classical ABCB1 and ABCG2 substrates. The interesting point is the demonstration of a direct ATPase activity inhibition and a reduction of MDR proteins transcription.
The major limit is its low therapeutic windows.
I have just few questions:
1. They stated that toxicity on non-cancer cell lines was lower than in cancer cell lines: did they tested ABCB1 and ABCG2 expression in normal cells?
2. They evaluated the reversal capacity on traditional drugs, but many recent target therapies, such as tyrosine kinase inhibitors are also ABCB1 and ABCG2 substrates. Did they have any idea in the potential efficacy in reversing resistance of TKIs?
Author Response
Response to Reviewer 1 Comments
The manuscript by Chao Guo ad coll. evaluated the efficacy of the new indole-derivative LS-2-3j in reversing multidrug resistance mediated by ABCB1 and ABCG2 in three highly resistant cell lines, in their sensitive parental counterpart and in three “non-cancer” cell lines. They observed high activity of the compound in increasing the cell toxicity as well as cell uptake and efflux inhibition of classical ABCB1 and ABCG2 substrates. The interesting point is the demonstration of a direct ATPase activity inhibition and a reduction of MDR proteins transcription.
The major limit is its low therapeutic windows.
I have just few questions:
Point 1: They stated that toxicity on non-cancer cell lines was lower than in cancer cell lines: did they test ABCB1 and ABCG2 expression in normal cells?
Response 1: Expression of ABCB1 and ABCG2 was not tested in normal cells in this study,though some research showed ABC transporters were expressed in the small intestine, colon and liver tissues. On the one hand, I don’t think the cytotoxicity of LS-2-3j is related to expression level of ABCB1 and ABCG2 in cancer or non-cancer cells. On the other hand, the purpose of LS-2-3j is to increase sensitive effect of chemotherapeutic drugs on tumors by means of combination therapy. So, I think the safety of the chemotherapy drugs used in further clinical therapy is more important. Meanwhile, as leading compound, further structure optimization studies of LS-2-3j to decrease toxicity are warranted.
Point 2: They evaluated the reversal capacity on traditional drugs, but many recent target therapies, such as tyrosine kinase inhibitors are also ABCB1 and ABCG2 substrates. Did they have any idea in the potential efficacy in reversing resistance of TKIs?
Response 2: Thanks for your suggestion. We did not perform experiments to confirm the potential efficacy of LS-2-3j in reversing resistance of TKIs. Thus, we cannot draw any conclusion. To our knowledge, several TKIs were found to interact with the major MDR transporters, such as ABCB1 and ABCG2. TKIs such as gefitinib, erlotinib, vandetanib and lapatinib have been shown to inhibit ABCB1 and ABCG2 function. Subsequently, the TKIs can attenuate downstream signalling pathway involved in cancer proliferation, invasion, metastasis and angiogenesis by inhibiting downstream signal molecules such as signal transducers and activators of transcription, protein kinase B/AKT, ERK1/2 and so on. LS-2-3j, a new chemically synthesized indole compound, is detected to inhibit the function and down-regulated the expressing of ABCB1 and ABCG2 in this manuscript, which suggested that LS-2-3j may have the potential efficacy in reversing resistance of TKIs, and further related studies are warranted.
Reviewer 2 Report
The work by Li and co-workers is an interesting example of rational design of a biologically active molecule inspired by related structures. Although the authors honestly also reports that still a some errors are present in the text and some of the results are not reported in a suitable way. These are my main suggestions:
-Abstract: I would change the first phrase from " 10-oxo-5-(3-(pyrrolidin-1-yl) propyl)-5,10-dihydroindeno [1,2-b] indol-9-yl propionate (LS-2-3j), a new chemically synthesized indole compound, has some analogues known to be..." to "10-oxo-5-(3-(pyrrolidin-1-yl) propyl)-5,10-dihydroindeno [1,2-b] indol-9-yl propionate (LS-2-3j) is a new chemically synthesized indole compound and some related analogues are known to be..."
I would furthermore remove the last phrase of the abstract, conclusions should be only in the conclusion paragraph and the abstract should give just the general informations about the article and not considerations or evaluations from the author.
-Introduction: in line 45 I would use "cause" rather than "driver", in line 56 it would be better to use "Furthermore" is substitution of "What's more". in line 57 it would be better tu use "put" or "made" instead of "gone". in line 59 I would change the phrase "is a novel synthetic small molecule indole compound synthesized by" to "is a novel indole-based small molecule synthesized by" to avoid the double use of synthetic/synthesized.
-Results: in line 68, 75, are reported two percentage values that are not precise, in scientific literature is not correct to report something "was nearly" or "more than" the correct and specific percentage should be reported.
-Discussion: in line 188 it is better to change "important" to "main". in line 204-205 there is still reported a percentage value not precise, it should be reported and discussed with the exact value.
The conclusion in line 249 should be changed from "synthesized indole compound, have not been reported about multidrug resistant reversal effects." to " synthesized indole compound reported for the first time, has multidrug resistant reversal effects." and in line 252 "explore" should be changed to "discovery".
-Materials and Methods: the paragraph about the synthesis is not written in the correct way and a lot of data is missing to make the work reproducible. In a procedure it is ok to report the equivalents of a compound used but should be reported also the amount of solvent used per mmol of the limiting reagent, to have a uniform experimental part it would be better always to give the mol or mmol of each reagent used and the amount of solvent. this is missing in the first procedure. In this one are missing also two important things, if you purify the compound using a determined solvent system you should report also the ratio of the solvents used or the gradient realized for the purification, also Yield of the reaction is missing.
In the second chemical procedure the author is presenting the use of ml of a solution in relation to equivalents used, this is not correct and also in this case reagents should be reported in mmol, the volume of pyridine used is still missing, the reaction time is missing and also the information about the purification (only solvent system reported and not the ratio or gradient), Yield of the reaction is missing and the worst thing is that the characterization of the compound is totally missing.
In the third and last chemical procedure the reagents should be reported in mmol and the volume of pyridine is missing, furthermore acyl chloride has been used so "sulfonic chloride" should be removed. Also in this procedure are missing important information for the purification (solvents ratio or gradient) and the yield of the reaction is missing.
The 13C spectra of this last compound should be checked because it seems that a carbon atom is missing, the total carbons of the molecule are 25, with 2 symmetrical carbons, the are are all different so there should be 23 signals and only 22 signals are reported.
Furthermore in all the 3 procedures CDCl3 should be reported with 3 as subscript.
Figure 7 is not immediate and representative for many reasons, first compound 1 should be reported in a complete way and not with abbreviations because you are reporting a unique compound and not some analogues with different substituents. In all the reaction arrows should be reported reaction time and reaction temperature and yield of each chemical passage.
The tests on the molecule seems correct and accurate but some important chemical data is missing, there are just 3 molecules synthesized in this work and it is not acceptable that the characterization data is missing, H and C NMR experiments for the characterization of LS-2-2a should be done with mass analysis. The C NMR data about LS-2-3j is not correct. Since this data is missing I suggest to accept this work with major revisions.
Author Response
Response to Reviewer 2 Comments
The work by Li and co-workers is an interesting example of rational design of a biologically active molecule inspired by related structures. Although the authors honestly also reports that still a some errors are present in the text and some of the results are not reported in a suitable way. These are my main suggestions:
-Abstract: I would change the first phrase from " 10-oxo-5-(3-(pyrrolidin-1-yl) propyl)-5,10-dihydroindeno [1,2-b] indol-9-yl propionate (LS-2-3j), a new chemically synthesized indole compound, has some analogues known to be..." to "10-oxo-5-(3-(pyrrolidin-1-yl) propyl)-5,10-dihydroindeno [1,2-b] indol-9-yl propionate (LS-2-3j) is a new chemically synthesized indole compound and some related analogues are known to be..."
I would furthermore remove the last phrase of the abstract, conclusions should be only in the conclusion paragraph and the abstract should give just the general information about the article and not considerations or evaluations from the author.
-Introduction: in line 45 I would use "cause" rather than "driver", in line 56 it would be better to use "Furthermore" is substitution of "What's more". In line 57 it would be better to use "put" or "made" instead of "gone". In line 59 I would change the phrase "is a novel synthetic small molecule indole compound synthesized by" to "is a novel indole-based small molecule synthesized by" to avoid the double use of synthetic/synthesized.
-Results: in line 68, 75, are reported two percentage values that are not precise, in scientific literature is not correct to report something "was nearly" or "more than" the correct and specific percentage should be reported.
-Discussion: in line 188 it is better to change "important" to "main". in line 204-205 there is still reported a percentage value not precise, it should be reported and discussed with the exact value.
The conclusion in line 249 should be changed from "synthesized indole compound, have not been reported about multidrug resistant reversal effects." to” synthesized indole compound reported for the first time, has multidrug resistant reversal effects." and in line 252 "explore" should be changed to "discovery".
-Materials and Methods: the paragraph about the synthesis is not written in the correct way and a lot of data is missing to make the work reproducible. In a procedure it is ok to report the equivalents of a compound used but should be reported also the amount of solvent used per mmol of the limiting reagent, to have a uniform experimental part it would be better always to give the mol or mmol of each reagent used and the amount of solvent. This is missing in the first procedure. In this one are missing also two important things, if you purify the compound using a determined solvent system you should report also the ratio of the solvents used or the gradient realized for the purification, also Yield of the reaction is missing.
In the second chemical procedure the author is presenting the use of ml of a solution in relation to equivalents used, this is not correct and also in this case reagents should be reported in mmol, the volume of pyridine used is still missing, the reaction time is missing and also the information about the purification (only solvent system reported and not the ratio or gradient), Yield of the reaction is missing and the worst thing is that the characterization of the compound is totally missing.
In the third and last chemical procedure the reagents should be reported in mmol and the volume of pyridine is missing, furthermore acyl chloride has been used so "sulfonic chloride" should be removed. Also in this procedure are missing important information for the purification (solvents ratio or gradient) and the yield of the reaction is missing.
The 13C spectra of this last compound should be checked because it seems that a carbon atom is missing, the total carbons of the molecule are 25, with 2 symmetrical carbons, there are all different so there should be 23 signals and only 22 signals are reported.
Furthermore in all the 3 procedures CDCl3 should be reported with 3 as subscript.
Figure 7 is not immediate and representative for many reasons; first compound 1 should be reported in a complete way and not with abbreviations because you are reporting a unique compound and not some analogues with different substituents. In all the reaction arrows should be reported reaction time and reaction temperature and yield of each chemical passage.
The tests on the molecule seems correct and accurate but some important chemical data is missing, there are just 3 molecules synthesized in this work and it is not acceptable that the characterization data is missing, H and C NMR experiments for the characterization of LS-2-2a should be done with mass analysis. The C NMR data about LS-2-3j is not correct. Since this data is missing I suggest accepting this work with major revisions.
Response: Thanks for your detailed advices. We have corrected in revised manuscript according all your comments. For the chemical synthesis part, we have added missing information of the three chemical procedures. What’s more, Yield and time of the reaction and the characterization of the compound LS-2-2a were added to the manuscripts. And we have revised the C NMR data about LS-2-3j.
Reviewer 3 Report
This is a well-organized manuscript and the scientific methods are sound. I have only three comments.
The bands in Figure 6C are very faint. Why was so little protein loaded on the gel, compared to the run shown in Figure 6D? Could a Western with stronger bands, more similar to the one shown in Figure 6D, be displayed in Figure 6C?
2. There is a vast literature now on ABC transporter inhibitors developed for the putative use of re-sensitizing resistant cancer cells. However, not one of these agents has been approved for clinical use. The authors should comment on the hurdles specifically preventing ABC transporter inhibitors' clinical utility. For example, specific tumor targeting, perhaps with the use of nanotechnology, may be needed to address toxicity issues. Also, selection of specific tumors with particularly high expression of a particular ABC transporter would increase the chances of success of this approach. These points should be added to the discussion.
3. Finally, as indicated above, extensive editing of the manuscript is required, as it is full of grammatical errors, too many for me to list here.
Author Response
Response to Reviewer 3 Comments
This is a well-organized manuscript and the scientific methods are sound. I have only three comments.
Point 1: The bands in Figure 6C are very faint. Why was so little protein loaded on the gel, compared to the run shown in Figure 6D? Could a Western with stronger bands, more similar to the one shown in Figure 6D, be displayed in Figure 6C?
Response 1: We do this experiment several times and we chose a better picture as shown in revised Fig. 6C. So picture with stronger bands has been displayed in Figure 6C.
Point 2: There is a vast literature now on ABC transporter inhibitors developed for the putative use of re-sensitizing resistant cancer cells. However, not one of these agents has been approved for clinical use. The authors should comment on the hurdles specifically preventing ABC transporter inhibitors' clinical utility. For example, specific tumor targeting, perhaps with the use of nanotechnology, may be needed to address toxicity issues. Also, selection of specific tumors with particularly high expression of a particular ABC transporter would increase the chances of success of this approach. These points should be added to the discussion.
Response 2: Thanks for your suggestions. The points of the hurdles specifically preventing ABC transporter inhibitors' clinical utility were added to the discussion in the manuscript (line249-266).
Point 3: Finally, as indicated above, extensive editing of the manuscript is required, as it is full of grammatical errors, too many for me to list here.
Response 3: We have carefully checked the manuscript. All the errors of grammar and spelling have been corrected accordingly in the revised manuscript. And we also submitted the revised manuscript to MDPI English editing service.
Round 2
Reviewer 2 Report
The Author corrected language errors, figure errors and completed the experimental data that now presented in the right way. I checked the paper and it now works really better, I think it is suitable for publication.